# Patient and Family-Centered Care for Pediatric Intraluminal Pulmonary Vein Stenosis: Case of a 3 Year Old Patient with Focus on Nurse Practitioner Role

**DOI:** 10.3390/children8070567

**Published:** 2021-07-01

**Authors:** Christina M. Ireland, Ryan Callahan, Kathy J. Jenkins

**Affiliations:** 1Department of Cardiology, Boston Children’s Hospital, Boston, MA 02115, USA; Ryan.Callahan@cardio.chboston.org (R.C.); Kathy.jenkins@childrens.harvard.edu (K.J.J.); 2Department of Pediatrics, Harvard Medical School, Boston, MA 02115, USA

**Keywords:** pulmonary vein stenosis, nurse practitioner, management

## Abstract

A nurse practitioner’s experience in managing children with intraluminal pulmonary vein stenosis. A case study of a 3-year-old patient with multi–vessel intraluminal pulmonary vein stenosis.

## 1. Introduction

Intraluminal Pulmonary Vein Stenosis (PVS) is a rare disorder which involves progressive narrowing of the pulmonary veins and can lead to pulmonary hypertension, right heart failure and death [1]. PVS can occur in young infants independently, in those with chronic lung disease and more commonly in conjunction with complex congenital heart disease. Managing this disease is unlike other cardiac defects because reoccurrence of the stenosis is expected due to the mechanism of the disease and is not completely understood or easily treatable. Although the pathophysiology of PVS is similar among patients, each patient and family brings a unique set of circumstances, and no one patient is the same [2]. While our team offers state-of-the-art multimodality treatment, provision of patient and family-centered care requires much more than medical therapy and intervention. To highlight the experience of families of PVS, as well as the support provided by and unique perspective of a nurse practitioner with a decade of experience in clinical and family support for PVS, we provide an illustrative case report and subsequent discussion of relevant issues. Parental permission was provided, and case reports at our institution are not considered research and are exempt from IRB approval.

### A Case of a 3 Year Old with PVS

A 3 year old ex 25 week premature twin with a history of secundum atrial septal defect was diagnosed with multivessel PVS.

They also had a history of bronchopulmonary dysplasia and anemia of prematurity. They were presented at their local hospital surgical conference at age 5 months upon the newly diagnosed PVS. The hospital deemed them not to be a surgical candidate, and the family sought a second opinion. The lung scan performed pre-catheterization is shown in Figure 1.

The patient had a cardiac catheterization at 6 months of age at our institution which showed bilateral upper pulmonary vein involvement. They had repair of their pulmonary veins which consisted of ostial resection and reimplantation of the left upper pulmonary vein (LUPV) to the left atrial appendage with anterior pulmonary homograft patch augmentation, left atrial enlargement with anterior patch augmentation of the right upper pulmonary vein (RUPV) and right middle pulmonary vein (RMPV) and subtotal ASD closure with a 3 mm fenestration. They were started on imatinib at postoperative day 8, once they had tolerated feeds and their surgical incision was well healed. They also underwent gastrostomy tube insertion for nutritional supplement. They were discharged back home with the plan to obtain an echocardiogram and lung scan monthly.

They had a cardiac catheterization at an outside hospital performed 12 weeks later for reoccurrence of RUPV, RMPV, LUPV and lingula veins and new disease in the left lower pulmonary vein (LLPV). Because the LLPV was unable to be accessed at the local hospital, they returned to Boston three weeks later for balloon dilation of the RUPV and LUPV and stent implantation in the lingula vein and LLPV.

They continued to require serial cardiac catheterizations but they were able to be spaced out to an average of every 4.5 months, while continuing imatinib during the next 2 years. They were making developmental strives and nutritional gains, although they still were behind compared to their fraternal twin.

When they were 2 ½ years old, inability to fracture the LLPV stent was encountered, and after 2 failed attempts in the catheterization lab, they were brought to the operating room for removal of the stents. The operation consisted of opening of the stent and patch plasty of the LUPV and LLPV along with fenestrated ASD closure with 2 mm fenestration with pulmonary homograft.

A planned 8-week postoperative cardiac catheterization was obtained, and the angiographic findings were significant for recurrent obstruction of all the pulmonary veins including some of the segmental branches at the end of the surgical patch material. All pulmonary veins were serially dilated using conventional and cutting balloons.

They were discharged home the following day with the plan to continue imatinib and have monthly surveillance testing conducted locally. They had a planned catheterization scheduled for 12 weeks.

## 2. Discussion

This case study is an example of a child with a history of prematurity who presented multivessel PVS along with a shunt lesion. They are being treated with multimodal management, including surgical and catheter-based interventions and targeted inhibition of myofibroblast proliferation with imatinib. They are now an active toddler weaned off oxygen and continue on gastrostomy tube supplemental feeds. They have had nine cardiac catheterizations and two cardiac surgeries to intervene on pulmonary veins. They continue to be monitored closely with echocardiograms and lung scans as indicated. Their lung scan post catheterization #9 is shown in Figure 2. They have overall tolerated imatinib, but their growth has been slow.

Their story continues, as it does for many of patients with PVS, and they will continue to need medical care and support, awaiting the next chapter. There have been many advancements in medical and surgical management of children with PVS over the past decade. That said there are still many unknowns of this lethal disease. Having a framework of how to care for these children and their families embodying principles of patient and family-centered care including a dedicated full time nurse practitioner has shown to be invaluable in the management of patients with PVS. The Family Management Style Framework (FMSF) is a nursing framework that describes the process of family management, how the individual family defines the condition, management of the disease and perceived consequences of the condition [3]. Implementing this framework in managing this unique PVS population on a daily basis is the basis of the nurse practitioner’s role.

Implementing the Family Management Style Framework (FMSF) framework for this unique population on a daily basis is the nurse practitioner’s role Although PVS is rare, our team has cared for over 200 patients. In most circumstances, the first person that the family has contact with is the nurse practitioner. When a new patient’s parent calls as in the case study, validating their emotions and the journey that they are going through is what first occurs. Explaining that PVS is a lethal and aggressive disease and that despite our best efforts, we may not win the fight is an important step. Being upfront on the aggressiveness of this disease, along with providing hope when giving a second opinion, is one strategy that has to be balanced. Offering a second opinion that may completely differ from the referring hospitals is a situation that occurs frequently. It is important to lay out a treatment strategy to the family that outlines the need of pulmonary vein assessment, pulmonary vein intervention, medical treatment and pulmonary vein surveillance are all covered in the first conversations with the family.

### 2.1. Multidimensional Assessment and Treatment

When performing the pulmonary vein assessment on a new patient, assessing the clinical status, evaluating each pulmonary vein, and also determining the aggressiveness of the disease are the first steps. The age of the child can play an important role in estimating the aggressiveness of the disease. Children that are four months and younger are at an increased risk of a poor outcome [4].

Pulmonary vein assessment is performed with echocardiogram to assess for congenital heart disease including left-to-right shunts, pulmonary vein anatomy and gradients, right ventricular pressure and right ventricular function. A nuclear lung perfusion scan is extremely helpful by demonstrating the distribution of blood flow to each lung segment. We use both of these modalities when diagnosing PVS and monitoring for disease reoccurrence. Cardiac catheterization is then performed to assess hemodynamics and perform a comprehensive pulmonary vein assessment, followed by pulmonary vein interventions. Chest CT is also useful in delineating the pulmonary vein anatomy and disease severity as well as for characterizing lung disease.

Following the initial assessment, patients typically undergo surgical repair. Over the years, many different surgical techniques have been tried in order to optimize the pulmonary veins. If a significant hemodynamic left-to-right lesion is also present, such as a ventricular septal defect or atrial septal defect (ASD), this would also be addressed at the time of surgical repair. The main goal of surgery is to relieve the obstruction and change the geometry of the veins in an effort to encourage laminar flow [5,6].

### 2.2. Managing Quality of Life versus Longevity

During the first admission, the Palliative Care team (PACT) is consulted. Their team consists of a physician, nurse practitioner and social worker who meet with the family in order to provide support and to establish goals of care. During the course of the child’s hospitalizations, the PACT team provides ongoing support, and goals are reassessed. This is extremely valuable for all healthcare team members; therefore, the goals of care are in line with medical treatment.

### 2.3. Managing Side Effects

Once the pulmonary veins have been intervened with, the next step is consideration of starting targeted medical therapy in order to reduce myofibroblast activity. At Boston Children’s Hospital we use imatinib (Gleevec^®^, Novartis Inc., Basel, Switzerland) as a primary therapy and add bevacizumab (Avastin^®^, Genentech Inc., South San Francisco, CA, USA) in patients with primary PVS or those who have failed monotherapy [7,8]. Imatinib is an oral chemotherapy that targets PDGFR–alpha and beta receptors (tyrosine kinase receptors) on the surface of myofibroblast-like cells. This drug is given daily by mouth or by nasogastric/gastric tube. The medication using the tablet formulation is one that has a bitter taste and high volume per dose. An oral suspension has been available since 2016, but finding a pharmacy that compounds this medication can be challenging [9]. Once the child is started on imatinib, it is not unusual to have daily conversations with the parents to help minimize side effects of the medication, such as feeding intolerance. This is usually confounded by the fact that patients with PVS already have feeding difficulties and manipulation of therapies should receive one factor at a time. Giving the drug with sleep is often better tolerated, as well as having a bit of food in their stomach. Children are usually given a premedication of ondansetron or lorazepam to help alleviate nausea and vomiting. In some children with a history of emesis prior to starting imatinib, these medications do not alleviate all of their symptoms.

### 2.4. Feeding and Nutrition

A majority of the patients have feeding issues which can be independent from their PVS and are multifactorial. Feeding issues include poor per oral intake, emesis, gastroesophageal reflux, aspiration, and poor weight gain. Many patients need to have their feeds supplemented with gastric or postpyloric tube feeds. Providing guidance to the family on how to manage the feeds can be an ongoing process and is trial and error. Transitioning from traditional bolus feeds to continuous feeds is one strategy. Achieving weight gain is imperative to PVS patients with the hope being that as the patient gains weight, the pulmonary veins grow. Involving the gastroenterology and feeding team at the first hospitalization is imperative to attempt to avoid aspiration and gastrointestinal reflux.

### 2.5. Communication and Care Continuity

Continuity of care is one concept that is embedded in the role of the nurse practitioner, caring for these vulnerable patients. The ability to be in close contact with families has been instrumental in being able to react swiftly to symptoms of reoccurrence. Symptoms of pulmonary vein restenosis typically consist of increased work of breathing, feeding intolerance and irritability. Keying into the unique symptom or symptoms manifested by each child when they experience recurrence is important to avoid a delay in intervening to treat pulmonary vein stenosis recurrence. As mentioned prior, daily conversations may occur with the families of newly diagnosed cases and those recently discharged from hospital settings. Weekly follow-up via phone is continued while the patient is considered to have aggressive disease.

### 2.6. Monitoring for PVS Recurrence

Recurrence of PVS can be silent. Monthly echocardiograms and lung scans in the first few months of diagnosis is imperative in order to monitor the pulmonary vein status. Performed together, the two tests resolve the limitations that can occur if performed independent of one another. After surveillance testing is performed, the findings are reviewed with families, they are reminded of the signs and symptoms to monitor for, and the drug therapy is dose-adjusted based on body surface area.

If surveillance testing from echocardiogram and/or lung scan shows possible recurrence, the next step is usually going to the cardiac catheterization lab for evaluation and possible interventions: angiography, intravascular ultrasound, conventional/cutting balloon angioplasty and possible placement of stents [10]. Each pulmonary vein is assessed and intervened with if needed. Determining if the stenosis is proximal or distal, recurrence or progression is important information to be learned from the catheterization. This information helps with understanding the aggressiveness of the pulmonary vein stenosis. Cardiac catheterization is the predominant intervention used to treat restenosis of the pulmonary veins in the long term. Patients who respond to therapy do not require re-intervention or their interval between transcatheter reintervention increases with time.

Bloodwork is conducted initially every two weeks and then spread out over time to every 3 months monitoring for drug toxicity of imatinib. Reviewing the lab work with the family and letting parents know if the drug is affecting their child’s immunity and vital organs is very important. Most parents at the initiation of the medication are concerned with the possible side effects the medication may have on their child’s overall health. The majority of the patients have had only minor side effects from the drug therapy. Reviewing the lab test results with parents and emphasizing when tests are in normal or expected range can also give reassurance that their child is tolerating the medication.

### 2.7. Risk of Infection

Parents of PVS children keep their child protected against infectious agents as best they can. To some, it means avoiding daycare, limiting interaction with family and friends, and avoiding sick contacts. To most, it means daily worrying to keep their child free from viral or bacterial infections. Providing families support and guidance on how to avoid exposure to infections is accomplished through frequent phone calls. Many children are on oral immunosuppressants that prohibit them for receiving live vaccines while taken medications and six months after.

### 2.8. Case Management

Case management is a huge component of discharge planning from the hospital and care at home in the community. This includes working on prior authorizations for medications, feeding supplies, formula, oxygen, respiratory equipment, nursing support and medical testing (echoes and lung scans and procedures such as cardiac catheterization). Due to the aggressiveness of pulmonary vein stenosis, time is often of the essence. Dealing with the prior authorization and appealing the denial are extremely time sensitive and can monopolize plenty of clinical time of the provider. Looking ahead and identifying the medications, supplies and medical treatments that are needed for discharge as soon as they are started in the hospital is one way of assuring that discharge will not be delayed. Changing of insurance due to job changes, unemployment and local state assistance can cause many hours of advocating for these services. Much time is spent on writing letters to insurance companies or on the phone advocating for the necessity of having the medication as a compound. Many families have spent endless hours trying to navigate the healthcare system, which at times is just chaotic. The insurance system is not set up for pediatric issues such as needing medications compounded or time-sensitive treatment for conditions such as PVS.

### 2.9. Support for PVS Survivors

Children with PVS are living longer, but new challenges and opportunities occur. Children that had stents put in their pulmonary veins as infants may become hardware-limited and may face having surgery to remove the stents, with fear of reactivation of disease activity. Children with other congenital heart defects may also face procedures and uncertainty. The role of child life specialist is invaluable to help prepare children for upcoming surgeries and hospitalizations. Children who were once totally gastrostomy-tube dependent are able to be weaned from their supplement feeds. They are attending school and participating in out-of-school activities with their peers. Some of the older children (age 10–12 years) are figuring out which sports accommodate their limited physical abilities. Golf, baseball and dance are a few of the sports that children have thrived in.

## 3. Conclusions

There have been many advancements in medical and surgical management of children with PVS over the past decade. However, there are still many unknowns and challenges of this complex and potentially lethal disease. The Family Management Style Framework directing care for these children and having a dedicated full-time nurse practitioner has shown to be invaluable in providing patient and family-centered care for PVS.

## Figures and Tables

**Figure 1 children-08-00567-f001:**
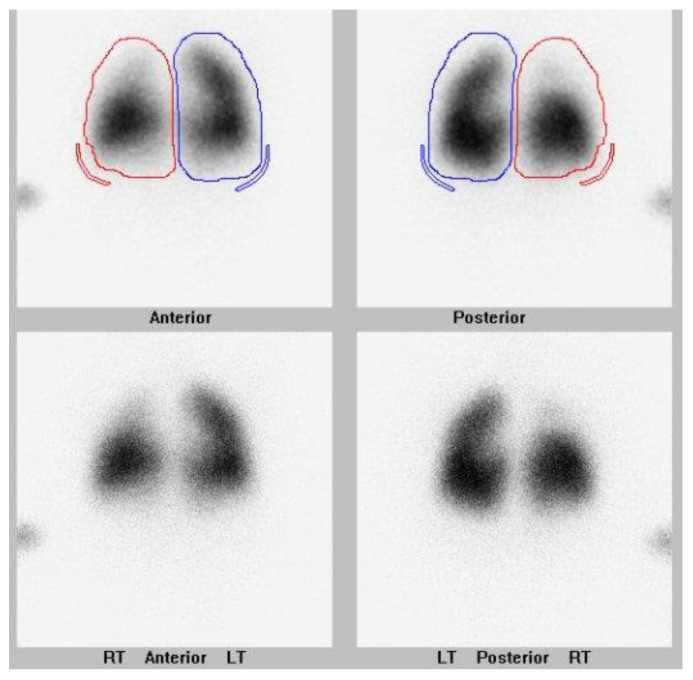
Nuclear lung perfusion scan of a 6 month old patient with a history of bilateral upper pulmonary vein stenosis. Image is pre-intervention. Differential pulmonary perfusion as quantitated based on Tc-99m MAA localization is 56% left and 44% right lung. There is decreased perfusion to the right upper lung. Right lung is outlined in red. Left lung is outlined in blue.

**Figure 2 children-08-00567-f002:**
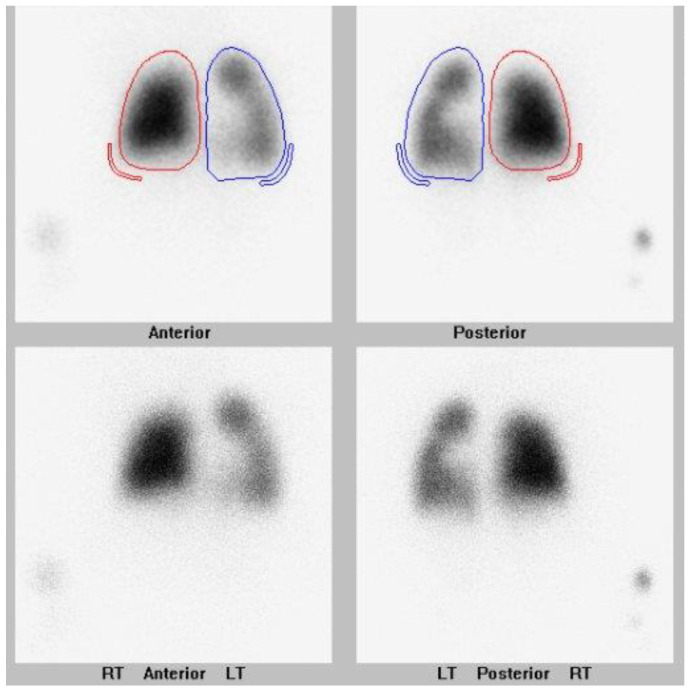
Most recent nuclear lung perfusion scan of patient who is now 3 years old with multivessel pulmonary vein stenosis involving all pulmonary veins. Patient has undergone 2 cardiac surgeries and 9 cardiac catheterizations on her pulmonary veins. Differential pulmonary perfusion as quantitated based on Tc-99m MAA localization is 34% left lung and 66% right lung. There is further diminished perfusion in the left compared to the right lung. Within the right lung, there is a similar pattern of diminished perfusion in the apex relative to the remainder of the lung. Within the left lung, there is interval slightly improved relative perfusion in the upper lung, and relatively diminished perfusion in the lower lung. Right lung is outlined in red. Left lung is outlined in blue.

## Data Availability

Not applicable.

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
