# Peer review of "Patient and Family-Centered Care for Pediatric Intraluminal Pulmonary Vein Stenosis: Case of a 3 Year Old Patient with Focus on Nurse Practitioner Role"

_children, 2021, doi:10.3390/children8070567_

Round 1

Reviewer 1 Report

The case report titled: << A Nurse Practitioner Experience in Managing Pediatric Intraluminal Pulmonary Vein Stenosis >> is authored by Christina M Ireland et al.

The authors describe how they managed to provide health care to the specific case of a 3-year-old infant with Pulmonary Vein Stenosis.

The topic is of great interest, the procedures are well described by the author, showing outstanding professionalism, however, some concerns need to be addressed before publication.

  1. Please re-write the abstract (<100) to be more descriptive and specific. In your abstract we could find:

- a short background (1 sentence) to contextualize what is PVS.

- how PVS is characterized in infants (1 sentence)

- Present the case and the specific issue. (1 sentence)

- what has been done to overcome and manage the specific case

- conclusion (1 sentences)

  1. Organize the main text in sections: 1-Introduction; 2-Case; 3-Discussion; 4-Conclusion

  1. The present introduction: paragraphs 2 and 3 are not necessary for the introduction as they are talking about one of the authors’ general practice. The introduction must focus on briefly describing the disease, the specific clinical challenges, and known therapeutic strategies.

Also, remove the paragraphs from :<< By having the opportunity …. (line 125) to … key form of support. (line 142).

This could be placed in the discussion section, avoiding using I, Me, … but please prefer using a passive form, and/or: We, our group, our team, etc. As you are the author, we understand that you did the job. Using a passing form in scientific reports will not minimize the importance of the work you are providing, but it may help other practitioners to relate to your description and improve their practice.

  1. Please replace the paragraph’s title << I would like to end…>> by << Case >>, or << Case of a 3-year-olkd PVS patient >>.

  1. Do not put the name of the patient on your manuscript. Please start your sentence by a passive form, i.e.:

<<A 3-year-olf patient ex 25-week premature twin … was diagnosed with multi-vessel PVS. >>

  1. Add a brief discussion section about PVS and the specific case reported.

You can organize your discussion as following:

  • Incidence of children with PVS worldwide, in your country, or in your hospital.
  • Cite 1 or 2 examples of reported advances in medical/surgical management of children with PVS. Add references. Sentence line 211 can introduce this part of the discussion. Add references.
  • Briefly discuss the management strategies related to the specific case. You can put here, some statements of the paragraphs 2 and 3 that the reviewer suggested to remove from introduction, and re-write them in a passive form, not using I, me, etc.
  1.  Conclude the paper with a Conclusion section about the management of children with PVS. (1 sentence)

Minor:

  1. Change the title for a more specific form related to the case you are describing:

i.e. Management of Pediatric Intraluminal Pulmonary Vein Stenosis: Case of a 3-year old patient.

  1. Keywords: Use technical terms related to the disease you are describing, and/or the material and methods used to manage it:

i.e. Pulmonary Vein Stenosis, Atrial Septal Defect, Cardiac Catheterization

  1. Figures: define the color codes (red and blue) you used in each figure legend.

Author Response

The case report titled: << A Nurse Practitioner Experience in Managing Pediatric Intraluminal
Pulmonary Vein Stenosis >> is authored by Christina M Ireland et al.

The authors describe how they managed to provide health care to the specific case of a 3-year-old
infant with Pulmonary Vein Stenosis.
The topic is of great interest, the procedures are well described by the author, showing outstanding
professionalism, however, some concerns need to be addressed before publication.
1. Please re-write the abstract (<100) to be more descriptive and specific. In your abstract we
could find:
- a short background (1 sentence) to contextualize what is PVS.
- how PVS is characterized in infants (1 sentence)
- Present the case and the specific issue. (1 sentence)
- what has been done to overcome and manage the specific case
- conclusion (1 sentences)
Abstract rewritten
2. Organize the main text in sections: 1-Introduction; 2-Case; 3-Discussion; 4-Conclusion
The paper was reorganized in sections as suggested by the reviewer. Thanks for the guidance on
reformatting.
3. The present introduction: paragraphs 2 and 3 are not necessary for the introduction as they
are talking about one of the authors’ general practice. The introduction must focus on briefly
describing the disease, the specific clinical challenges, and known therapeutic strategies.
The author is a nurse practitioner and one of the goals of the paper was to give perspective from that
role.
Also, remove the paragraphs from :<< By having the opportunity …. (line 125) to … key form of
support. (line 142).
Paragraohs were removed and some content was put in discussion section.
This could be placed in the discussion section, avoiding using I, Me, … but please prefer using a
passive form, and/or: We, our group, our team, etc. As you are the author, we understand that you
did the job. Using a passing form in scientific reports will not minimize the importance of the work you are providing, but it may help other practitioners to relate to your description and improve their
practice.
4. Please replace the paragraph’s title << I would like to end…>> by << Case >>, or << Case of
a 3-year-olkd PVS patient >>.
This was revised as suggested. I used they .
5. Do not put the name of the patient on your manuscript. Please start your sentence by a
passive form, i.e.:
<<A 3-year-olf patient ex 25-week premature twin … was diagnosed with multi-vessel PVS. >>
This was revised as suggested.
6. Add a brief discussion section about PVS and the specific case reported.
You can organize your discussion as following:
ï‚· Incidence of children with PVS worldwide, in your country, or in your hospital.
ï‚· Cite 1 or 2 examples of reported advances in medical/surgical management of children with PVS.
Add references. Sentence line 211 can introduce this part of the discussion. Add references.
ï‚· Briefly discuss the management strategies related to the specific case. You can put here, some
statements of the paragraphs 2 and 3 that the reviewer suggested to remove from introduction, and
re-write them in a passive form, not using I, me, etc.
7. Conclude the paper with a Conclusion section about the management of children with PVS.
(1 sentence)
Please see revised discussion section and conclusion.
Minor:
1. Change the title for a more specific form related to the case you are describing:
I would like to keep the focus of the title to reflect the nurse practitioner experience.
Patient and Family-Centered Care for Pediatric Intraluminal Pulmonary Vein Stenosis: Case of a 3
year old Patient with Focus on Nurse Practitioner Role
2. Keywords: Use technical terms related to the disease you are describing, and/or the material
and methods used to manage it:

i.e. Pulmonary Vein Stenosis, Atrial Septal Defect, Cardiac Catheterization
All key words were spelled out.
3. Figures: define the color codes (red and blue) you used in each figure legend.
Right lung is outlined in red. Left lung is outlined in blue. Figures updated to define color
code,

Reviewer 2 Report

Thank you for submitting your great efforts.

There are very few reports of nurse practitioner experiences for patients with pulmonary vein stenosis.

I have some comments and questions that authors may consider.

  • There is no mention of an IRB approval for this manuscript.
  • The authors should describe informed consent for the case study.
  • The authors should not provide the name of the case study.
  • Would you provide more details on the specific solutions for feeding issues, minimizing side effects of medications, insurance matters, and keeping their child viral/bacterial illness-free? These things are the most important part of this manuscript.
  • The authors described the involvement PACT; please describe in more detail the specific role of this team.

Author Response

Thank you for submitting your great efforts.
There are very few reports of nurse practitioner experiences for patients with pulmonary vein
stenosis.
I have some comments and questions that authors may consider.
ï‚· There is no mention of an IRB approval for this manuscript.
IRB was not needed from our institution due to less than 4 subjects were used.
ï‚· The authors should describe informed consent for the case study.
An informed consent form was obtained from the parent of the case study.
ï‚· The authors should not provide the name of the case study.
Name was omitted from the case study.
ï‚· Would you provide more details on the specific solutions for feeding issues, minimizing side effects of
medications, insurance matters, and keeping their child viral/bacterial illness-free? These things are
the most important part of this manuscript.
Please refer to section feeding and nutrition
Please refer to section managing side effects
Please refer to risk of infection

The authors described the involvement PACT; please describe in more detail the specific role of this
team.
Managing quality of life versus longevity

Round 2

Reviewer 1 Report

The authors answered all reviewer's comments. They significantly improved the manuscript. The ideas are well organized.

Reviewer 2 Report

The authors have been well revised according to the reviewer's opinion.